# Valorization of Coffee Husk in Ternary Bio-Composites: Synergistic Reinforcement of Bio-Epoxy/Polylactic Acid for Enhanced Mechanical and Physical Properties

**DOI:** 10.3390/polym17223013

**Published:** 2025-11-13

**Authors:** Isaác Molina-Sánchez, Carlos Moreno-Miranda, Rodny Peñafiel, Mayra Paredes-Escobar, Pilar Pazmiño-Miranda, Miguel Aldás, Estefanía Altamirano-Freire, Nelly Flores

**Affiliations:** 1Research and Development Directorate, Food and Biotechnology Faculty, Universidad Técnica de Ambato (UTA), Av. Los Chasquis y Río Payamino, Ambato 180206, Ecuador; cs.moreno@uta.edu.ec (C.M.-M.); ml.paredes@uta.edu.ec (M.P.-E.); ne.flores@uta.edu.ec (N.F.); 2Departamento de Ciencias Exactas, Universidad de las Fuerzas Armadas-ESPE, Av. General Rumiñahui, Sangolquí 171103, Ecuador; 3Biosynergia Research Group, Food and Biotechnology Faculty, Universidad Técnica de Ambato (UTA), Av. Los Chasquis y Río Payamino, Ambato 180206, Ecuador; rd.penafiel@uta.edu.ec (R.P.); nd.pazmino@uta.edu.ec (P.P.-M.); 4Centro de Investigaciones Aplicadas a Polímeros, Departamento de Ciencias de los Alimentos y Biotecnología, Escuela Politécnica Nacional, Ladrón de Guevara E11-253, Quito 170517, Ecuador; miguel.aldase@epn.edu.ec; 5Faculty of Biotechnology Engineering and NaturalResources, Universidad Politécnica Salesiana, Sede Quito, Campus El Girón, Av. 12 de Octubre N2422 y Wilson, Quito 170143, Ecuador; kaltamiranof@est.ups.edu.ec

**Keywords:** agro-waste valorization, Coffea arabica husk, biocomposites, polylactic acid, bio-based epoxy, matrix-filler interface

## Abstract

This study investigated Coffee arabica husk (CAH) as a reinforcing filler to create sustainable biocomposites from agro-industrial waste. The research explored the relationship between processing, structure, and properties using two matrices: polylactic acid (LA) and a bio-based epoxy resin (BER). We found that CAH incorporation increased the elastic modulus in all composites, with the stiffening effect being more significant in BER-based systems. However, filler inclusion dramatically reduced composite toughness. Our analysis showed that melt processing significantly reduced the CAH aspect ratio, with BER causing a more pronounced reduction than LA. Conversely, LA showed a greater tendency to fill the porous voids of the CAH particles. This work demonstrates the crucial interaction of filler, matrix, and processing on a composite’s final performance. These materials have shown promises for sustainable packaging and other technical applications.

## 1. Introduction

The overarching trend toward sustainable materials development has recently focused significant research attention on agro-fiber composites, which offer a compelling pathway for minimizing their ecological footprint. Crucially, the sub-class known as green composites—defined by the utilization of natural fibers within a polymer matrix—has become a particular focal point for its potential to replace conventional, non-biodegradable materials [1,2]. These materials offer a unique opportunity to valorize agricultural or food waste, transforming it into valuable products [3]. This approach yields several key advantages: reduced production costs [4,5,6], and a manufacturing process that poses negligible worker hazards. Beyond their numerous eco-sustainability benefits [7,8,9], incorporating natural fibers into polymer matrices can also improve specific mechanical properties [10], such as stiffness, enabling these biocomposites to displace petroleum-based composites effectively.

The incorporation of bio-based fibers and particulates as reinforcement for polymer systems has been extensively investigated in contemporary scholarship. According to the published literature, biocomposites can be fabricated utilizing lignocellulosic reinforcements sourced from agricultural waste streams, such as *Abelmoschus esculentus*, coconut fiber, *Ananas comosus*, and *Cynara cardunculus*, in addition to cultivated organic reinforcements like milled wood, *Linum usitatissimum*, *Cannabis sativa*, *Corchorus capsularis*, *Hibiscus cannabinus*, *Boehmeria nivea*, and *Agave sisalana* [11,12,13,14]. When compared to plain polymer matrices, these composites frequently show better physical properties. However, common to engineered polymer systems (especially those at the nanoscale), the magnitude of these upgrades is contingent upon a number of important factors, including (i) the starting components’ chemical-physical attributes [15], (ii) filler dispersion (and fiber orientation, if applicable) [16], and (iii) matrix/fiber adhesion [17].

Additionally, because of the inherent heterogeneities of raw lignocellulosic fillers, green composites may exhibit notable data scattering in contrast to composites that use synthetic or inorganic fillers [18]. In addition to differences in the particle’s geometric form, microstructure, and surface texture, each unit may possess varying amounts of lignin [19,20], cellulose [21], and xyloglucan, all contributing to overall material component variability [22]. Their contact and relationship with the polymer matrix are directly affected by these differences, which usually lead to varying capacities to transfer stresses.

Bio-based epoxy resin (BER) is currently attracting considerable attention as a preferred eco-friendly substitute for conventional synthetic thermosets [23]. This growing preference is notable because BER is being increasingly examined for its utility as a binding agent within biocomposite systems. Its renewability is a primary advantage, stemming from its derivation from biomass sources like vegetable oils, lignin, or other natural feedstocks [24,25]. However, it is crucial to note that ecological value is not a universal characteristic of all BERs and depends heavily on the specific chemical structure and curing agents employed [26]. Despite this, BERs often exhibit attractive physical properties such as good adhesion, chemical resistance, and thermal stability [27], rendering the material appropriate for a wide spectrum of uses, including building and biomedicine, coatings, and specialized containerization.

Another widely studied and commercially successful biopolymer is polylactic acid (LA). This compound, which is a bio-based thermoplastic aliphatic polyester, is synthesized from regenerative feedstocks like dextrose (glucose) from corn or sucrose from cane [28]. LA is highly regarded for its favorable structural performance, including its superior modulus and tensile strength, often surpassing that of other bio-based polymers [29]. While its toughness may be lower when compared to conventional polymers such as low-density polyethylene (LDPE), LA nonetheless exhibits favorable thermal resistance and processability. These attributes establish its viability across numerous sectors, most prominently in containerization, but also with growing use in fibrous materials, vehicular components, and even certain medical implements [30,31]. Its perishability under specific composting conditions is a significant environmental advantage, although its degradation rate can vary depending on the environment.

Within the context of this work, we evaluated the feasibility of using *Coffea arabica* husk (CAH) waste as a filler for renewable composites. CAH is a significant agricultural byproduct of coffee production in regions such as Ecuador [32]. *Coffea arabica* is a bush species cultivated for its coffee beans, and the husk represents a readily available and often underutilized agricultural residue [33]. Its abundance and potential for low-cost acquisition make it an attractive option for valorization, addressing disposal challenges, and promoting a circular economy.

For this investigation, two distinct particulate fractions of ground CAH were utilized. These were integrated into two separate biopolymeric systems, and were characterized by varying particle sizing, geometrical shape, and consequently, total surface area: polylactic acid (LA), known for its relative rigidity, and a bio-based epoxy resin (BER), chosen for its potential for good adhesion and different mechanical behavior compared to LA. The investigated composite systems are complex due to the unique morphology of the CAH filler, which exhibits a porous platelet architecture and inherent heterogeneity in composition and structure [34]. These characteristics significantly influence filler dispersion and interfacial adhesion, ultimately affecting the composite’s performance reliability.

A thorough investigation looked at the relationship between these systems’ fabrication, microstructure, and performance. This required considering several phenomena, including the reinforcement’s length-to-diameter ratio subsequent to compounding, polymeric breakdown, and the matrix’s ability to infiltrate the interstitial spaces of the reinforcement’s porous particles. Ultimately, a factorial approach was used to evaluate a pair of parameters of interest: toughness and elastic modulus. This method sought to quantify how the properties of these biocomposites were impacted by the kind of polymer, filler size, filler content, and their interactions. In order to fabricate materials with precise properties, it is essential to model comprehensive systems.

## 2. Materials and Methods

### 2.1. Materials and Preparation

Figure 1 shows the materials and preparation and describes the following steps. *Coffea arabica* husks (CAHs) used in this study were collected from coffee farms located in the coastal region of Guayas Province, Ecuador (approximate coordinates: 2°10′ South, 79°45′ West or −2.17, −79.75). After collection, the CAHs were purified, including washing, to remove impurities and grinding to reduce particle size. Subsequently, the material was sorted via sieving to isolate specific fractions and was then air-dried for a period of approximately 12 h to ensure complete dehydration. Following this desiccation step, the pulverized CAH was separated using a dual-sieve arrangement, yielding two principal grades: Fraction A, with a particle dimension range of 150 µm to 300 µm, and Fraction B, comprising particles between 75 µm and 150 µm. These CAH specimens exhibit an apparent density of 0.42 g/cm^3^ and a true density of 1.54 g/cm^3^, as determined via helium pycnometry (Thermo Electron Corporation’s Pycnomatic ATC; Atlanta, GA, USA).

Polylactic acid (LA) and a bio-based epoxy resin (BER) were selected for the matrices, with their main characteristics detailed in Table 1. The LA used, an extrusion grade 2002D supplied by Nature Works, exhibits a D-lactic acid monomer content of approximately 4%. The BER employed in this research was a commercially available bio-based epoxy resin derived from epoxidized soybean oil and provided by WEGO Chemicals. This specific BER shows an approximate epoxy equivalent weight (EEW) of 200–250 g/eq and a viscosity of 500–800 cP at 25 °C. Its curing was performed using a polyamine hardener, maintaining a resin: hardener ratio of 2:1 by weight. According to the supplier’s technical datasheet, the bio-based content of this resin is estimated to be 60%. In general terms, Bio-based Epoxy Resins (BERs) are defined as epoxy polymers where a substantial fraction of their constituent monomers are sourced from regenerative biological supplies, for instance, lipid matter from plants, lignin, resin, or other biomass. The chemical architecture and distinctive attributes of BERs are highly dependent on the bio-based feedstock and the synthesis process employed. Consequently, the BER utilized in this work is characterized by its vegetable oil origin, EEW range, moderate viscosity, and amine-based curing system. This BER can be accurately described as a multiphasic material composed of complex biopolymers with distinct chemical structures and properties.

To mitigate the potential for hydrolysis, every constituent was subjected to vacuum desiccation for a full night at 90 °C prior to subsequent handling. Melt compounding was used in a Brabender internal mixer to create the biocomposites. This procedure was carried out for six minutes at a temperature of 190 °C and a rotor speed of 64 rpm. Table 2 lists the precise formulas that were employed. A Carver laboratory press was used to compress each batch of material after it had been melted and crushed into pellets. This molding process took place for two minutes at 190 °C and 180 bar. In order to create specimens with the proper geometry for further characterizations, the molded material was finally sliced.

### 2.2. Characterization Techniques

Figure 2 Shows a summary scheme of the characterization techniques used in this work.

#### 2.2.1. Characterization Techniques

A helium pycnometer (Thermo Electron Corporation’s Pycnomatic ATC, Atlanta, GA, USA) was employed to determine the true density of the CAHs. For each specimen, a set of ten readings was executed, and the resulting mean value was subsequently documented. In every instance, the scatter remained consistently below 0.01 g/cm^3^. Porosity was calculated using Equation (1).(1)$$Porosity\;\left(\%\right)=\left(1-\frac{\rho_{a}}{\rho_{r}}\right)\times 100$$
where $\rho_{a}$ and $\rho_{r}$ represent the CAHs bulk density and true density, in that order.

The proportion of CAH reinforcement occupied by LA (i.e., the intraphase ratio) was calculated utilizing Equation (2):(2)$$intraphase\;\left(\%\right)=\;\frac{\rho_{real}-\rho_{unfilled}}{\rho_{filled}-\rho_{unfilled}}\times 100$$

Here, $\rho_{real}$ denotes the composite material’s density. The parameters $\rho_{filled}$ and $\rho_{unfilled}$ relate, in order, to the maximal theoretical density of the blend (assuming full occupation of all CAH reinforcement voids by LA) and the minimal theoretical density (assuming no occupation of CAH voids by LA). Both $\rho_{filled}$ and $\rho_{unfilled}$ were established employing Equations (3) and (4), as follows:(3)$$\rho_{filled}=\;\rho_{CAH-pyc}\;\Phi_{CAH-pyc}+\;\rho_{PLA}\;\Phi_{PLA}$$(4)$$\;\;\;\;\rho_{unfilled}=\rho_{CAH-app}\;\Phi_{CAH-app}+\rho_{PLA}\;\Phi_{PLA}$$

Here $\rho_{CAH-pyc}$ denotes the density of CAHs, as determined by pycnometer (i.e., 1.54 g cm^−3^). In this context, $\Phi_{CAH-pyc}$ represents the volume proportion of the reinforcement within the material systems, using that aforementioned true density value for its calculation. Conversely, $\rho_{CAH-app}$ refers to the apparent density of CAH (i.e., 0.42 g cm^−3^), and $\Phi_{CAH-app}$ designates the volume share of CAH in the final compound, which is calculated based on this apparent density.

#### 2.2.2. Spectroscopic Analysis

Fourier-transform infrared spectroscopy with attenuated total reflectance (FTIR/ATR) was conducted using a Perkin Elmer spectrometer (PerkinElmer Inc., Shelton, CT, USA, operating across the wavenumber range of 4000–400 cm^−1^. Surface wettability was evaluated by measuring the liquid-substrate contact angle (WCA) using an FTA 1000 apparatus (First Ten Ångstroms, Newbury, Berkshire, UK) under ambient conditions. The procedure utilized an automatic liquid drop dosing system to deliver a 4 µL droplet of distilled water onto the polymer surface for each trial. Images of the sessile drop were captured 20 s after deposition. To ensure statistical reliability, the mean WCA readings were calculated and presented following examination of a minimum of five distinct locations per specimen.

#### 2.2.3. Morphological Examination of Composites and Fillers

Using ImageJ (Fiji (ImageJ2))as the image processing program, the key geometrical characteristics of the CAH particles—namely, their length, breadth, and thickness—were measured. Additionally, following their Soxhlet extraction in THF from the composites, the form parameters of CAHs were evaluated. The purpose of this extraction was to assess the possible impact of processing on filler size. For every sample, 90 particles were subjected to image analysis. Equation (5) was used to obtain the particles’ equivalent diameter, and Equation (6) was used to determine the filler aspect ratio ($A_{f}$).

Here, l_w_ and t correspond to the arithmetic mean of the dimension readings for the length, width, and thickness of the analyzed specimens, in that order.(5)$$D_{eq}=\sqrt{\frac{4lw}{\pi }}$$(6)$$A_{f}=\frac{D_{eq}}{t}$$

#### 2.2.4. Rheological Characterization

Both the melt and solution viscosities were measured in order to characterize the samples rheologically. Melt rheology measurements were made using a plate-plate rotating rheometer (Mars, Thermofisher, Waltham, MA, USA) on disk-shaped specimens that had been vacuum-dried overnight at 80 °C prior to testing. These tests, which included a frequency range of 1–100 rad/s, were carried out in frequency sweep mode at 190 °C with a constant strain of 5%.

Although the Mark–Houwink constants are typically employed to derive the viscosity-average molecular mass (Mv) of LA based on its inherent viscosity, this approach is unsuitable for BER due to its ambiguous chemical formulation. Consequently, Equation (7) was applied to calculate the proportional viscosity ($\eta_{rel}$) readings for all specimens. This involved standardizing the elution time of each specimen ($t_{F,c}$) against that of its neat counterpart matrix ($t_{F,m}$) to assess the loss of integrity in the matrices subsequent to fabrication.(7)$$\eta_{rel}=\frac{t_{F,c}}{t_{F,m}}$$

#### 2.2.5. Mechanical Characterization

Uniaxial tensile testing was executed following the ASTM D638-10 standard [35]. Eight replicate dumbbell-shaped test pieces were evaluated for every material batch using a traverse rate of 5 mm/min. The Young’s modulus (E) was defined as the preliminary gradient of each stress–strain curve, whereas energy absorption capacity (toughness) was quantified by the area beneath the stress–strain curve.

#### 2.2.6. Statistical Analysis

Following the procedures outlined by Baron et al. [36], a two-level complete factorial design was utilized to investigate the outcomes of altering the operational parameters. This methodology facilitates the examination of how diverse operating factors influence specific target attributes, with emphasis on each variable’s contribution, optimal pairing, and potential mutual interactions. Selecting a high and low value for every k variable ($x_{1},\;x_{2}$, …, $\;x_{n}$) permits the analysis and characterization of any observable characteristic, P($x_{1},\;x_{2}$, …, $\;x_{n}$), resulting in 2*^k^* combinations for experimental evaluation. Typically, a “Matrix of contrasts” provides a schematic summary of these pairings. Each row in this matrix specifies a unique set of conditions, and the columns aid in the calculation of each variable’s principal impacts and their interactions. A variable’s principal effect is defined as the disparity between the means (the mean associated with the variable’s superior level and the mean linked to its inferior level). Thus, the impact of each variable on the chosen property can be assessed while holding the other variables constant. For instance, the primary influence of the $x_{1}$ variable, $P_{x_{1}}$, quantifies the increase that the P property receives when the variable shifts from its inferior to its superior level. In statistical computations, the contrast matrix is populated with “+” and “−” symbols, based on the variable under consideration and its assumed superior or lowest setting. The numerical values that property P assumes for the $2^{k}$ sets of conditions are shown in the last column. It can be shown that the algebraic total of the values of P, each assigned the sign found in the respective column of the matrix, can be used to determine the major influence of any given variable “i.”(8)$$M.E.=\sum_{i=1}^{k}M_{i}$$Here i = (1,…k) represents the specific set of operating conditions ($k\;=\;2^{k}$), $M_{i}$ is the mean reading of the chosen characteristic. The aggregate is computed by adopting the arbitrary “$M_{i}$” reading as either additive or subtractive, based upon the symbol (“+” or “−”) it holds in the corresponding column of the contrast matrix.

However, identifying primary impacts alone is insufficient for a detailed design because it does not fully describe the impact that each variable has on the chosen attribute. Rather, it calls for determining the importance of all possible interactions between every variable. Multiplying the sign arrays of the corresponding variable columns in the contrast matrix generates the column that illustrates the interplay between parameters $x_{1}$ and $x_{2}$.

A variable’s primary effect is deemed statistically significant if its numerical value is at least two to three times greater than the standard error of the effect (SE), defined as the square root of the effect’s variance estimate. The calculation of the experimental error variance, denoted s^2^, is computed using Equation (9). This calculation uses $d_{i}$, the range of the property P between its maximum and minimum readings recorded for the generic i−th experimental condition.(9)$$s^{2}=\sum \left(s_{i}^{2}/g\right)=\sum (d_{i}^{2}/g)$$
where g denotes the count of degrees of freedom. The average variation of the effect (VE) is then calculated as Equation (10):(10)$$VE=(\frac{1}{g}+\frac{1}{g})s^{2}$$

Thus, the square root of this computed variance yields the error limit of the impact. Given that each ratio, effect/SE(effect), will conform to Student’s t distribution possessing  g degrees of freedom, this work adopted a more rigorous approach. Therefore, a *t*-test facilitates the assessment of the significance of each effect (pertaining to individual variables and their mutual interactions) at a given confidence level by selecting an appropriate confidence threshold (in this case, 95%).

## 3. Results

The physical form of the microparticles was evaluated via Scanning Electron Microscopy (SEM), with representative micrographs displayed in Figure 3A–D. This figure additionally includes a high-resolution image of the unprocessed husks and pulverized reinforcements. Panels A and B display a general perspective on the particle dimensions and configuration for CAH-A and CAH-B. Zambrano et al. [37], in their study, observe that these microparticulates are porous, parallelepiped-like structures possessing a constant thickness of 25 μm with mutable length and width. High-magnification images (Figure 3C,D) distinctly reveal the characteristic CAH structure, which incorporates several co-linear vascular fascicles. Surface impurities were also noted, a common observation for raw lignocellulosic fibers. A cross-sectional view of the CAHs substantiates the vascular fascicles possess a multi-sided cross-section and a central lumen of approximately 5 μm × 12 μm, which aligns with prior studies [34,36]. Based on a true density of 1.54 g cm^−3^ and a bulk density of 0.42 g cm^−3^, the porosity of the CAHs was calculated using Equation (2) to be approximately 73%.

Figure 4 illustrates the particle size distribution of CAH-A and CAH-B, explicitly showing the length (A), width (B), and thickness (C). A schematic representation of the geometric parameters measured for a typical CAH particle is included in panel D.

A comparison of the particle size distributions showed that CAH-A was consistently sized, with unimodal distributions for its length (L), width (W), and thickness (t). Conversely, CAH-B was notably more heterogeneous. Its length and width distributions were bimodal or multimodal, although its thickness distribution was narrow and constant, regardless of particle size. The average geometric parameters for both CAH types before melt processing are summarized in Table 3.

The disparity in size distribution resulted in a notable divergence in the initial aspect ratio. As calculated using Equations (5) and (6), the aspect ratio for CAH-B (12.4) was nearly double that of CAH-A (6.2). This observation is corroborated by the D_eq_ ratio, given that the characteristic dimension of both CAH-A and CAH-B microparticles was invariant.

In addition to physically defining the reinforcement, FTIR/ATR spectroscopy and WCA measurements were utilized to perform a first-stage physico-chemical assessment on both polymeric systems. The FTIR/ATR profiles for both LA and BER are presented in Figure 5.

According to data from the literature (e.g., [38]), the FTIR/ATR spectrum for the Lactic Acid (LA) component (interpreted as Polylactic Acid, PLA) is dominated by features characteristic of an aliphatic ester. The most significant finding is the powerful C=O stretching band at 1748 cm^−1^, which is the signature of the ester carbonyl group in the polymer backbone. Aliphatic C-H stretching vibrations for the methyl (CH3) groups are visible in the 2997 cm^−1^ (asymmetric) and 2946 cm^−1^ (symmetric) region. Further confirmation of the PLA structure is found in the fingerprint region, with prominent C-O and C-O-C stretching modes concentrated between 1225 cm^−1^ and 1093 cm^−1^. This pattern includes C=O bending at 1225 cm^−1^, and the key CH3 rocking modes at 956 cm^−1^ and 921 cm^−1^ that are often used to assess PLA’s amorphous state. The observation of a faint band at 3500 cm^−1^ confirms the presence of terminal O-H end groups, consistent with a polymer.

The FTIR spectrum for the Bio-based Epoxy Resin (BER), although of unknown precise composition, shows a clear structure consistent with a blend, specifically indicating the presence of a PBAT-like co-polyester and a starch component. The PBAT (poly(butylene adipate-co-terephthalate)) matrix is confirmed by its characteristic ester bands: the C=O carbonyl stretching at 1720 cm^−1^ and the C-O ester coupling band at 1274 cm^−1^. The polyester’s aliphatic and aromatic nature is demonstrated by the broad C-H stretching region (2950–2850 cm^−1^), and the phenylene groups from the terephthalate unit are supported by strong bands between 1574 cm^−1^ and 1019 cm^−1^. Critically, the strong band at 720 cm^−1^ acts as a diagnostic marker for the rocking mode of four or more adjacent CH_2_ groups, confirming the poly(butylene adipate) segment.

The starch-based component of the BER is confirmed by the broad O-H stretching vibration centered near 3300 cm^−1^, indicating the numerous hydroxyl groups present in the glucose units. This is further supported by complex vibrational modes between 1445 and 1325 cm^−1^. The concurrent identification of both polymer matrix functional groups and O-H groups in the BER and LA/PLA spectra is significant: the presence of multiple oxygenated moieties (esters, hydroxyls) in both polymers validates their potential for strong hydrogen-bonding interaction with any lignocellulosic or hydroxyl-rich fillers, which is a key conclusion for materials science analysis.

Wettability measurements showed a water contact angle (WCA) of 73° for LA and 83° for BER, indicating that LA is slightly more hydrophilic. Figure 6 shows SEM micrographs of all the LA- and BER-based composites. The images show good dispersion and adhesion in all samples, with slightly better adhesion observed in the BER-based composites. The micrographs also clearly show the different sizes and amounts of CAH particles within the polymer matrices.

The final aspect ratio was quantitatively assessed to determine the impact of processing on the filler, as it is a crucial parameter for composite material properties. Figure 7 illustrates the length distribution of CAH-A particles after extraction from LA (A) and CAH-B particles after extraction from BER (B).

A clear difference was observed between the two particle types. The CAH-A particles from the LA composite showed a unimodal distribution with a peak frequency at 150 µm and an average weight length of 170 µm. Conversely, the CAH-B particles from the BER composite had a bimodal distribution, with two distinct peaks at 100 µm and 325 µm, leading to a larger average weight length of 220 µm.

Both samples showed a bimodal length distribution with respect to the fillers that were taken out of the BER matrix. CAH-B particles displayed peaks at 100 µm and 325 µm, whereas CAH-A particles had maximum intensity occurred at 25 µm and 150 µm. Interestingly, compared to the CAH-A particles recovered from LA, which had a minimum length of 50 µm and 75 µm, respectively, very few CAH-B particles with a 25 µm length were detected in the BER matrix. CAH-A and CAH-B had average weight lengths of 115 µm and 205 µm, respectively. According to these findings, shrinkage was more noticeable after melt processing with BER than after processing with LA, especially for the CAH-A fillers.

Figure 7C illustrates the final aspect ratio (A_f_) as a function of processing time (Equations (5) and (6)) for both LA and BER composites containing CAH-A or CAH-B. CAH-B microparticles underwent a more significant size reduction, decreasing their Af from 12.43 to 7.27 (a 41% reduction) in LA-based composites and 12.43 to 6.48 (a 48% reduction) in BER-based composites. CAH-A fillers showed matrix-dependent behavior: in LA, the Af was retained mainly with only a 5% reduction, while in BER, a substantial reduction from 6.16 to 4.23 (a 31% reduction) was observed.

As shown by the SEM data, another investigation was carried out to examine the polymers’ capacity to enter the CAH particles’ porous channels. Three situations are depicted in Figure 8: The first theoretical scenario involves a CAH microparticulate that is completely devoid of polymer, exhibit 73% porosity and a reinforcement density equivalent to its bulk density. The second hypothetical case presents a CAH microparticulate that is saturated with the polymer, where the reinforcement density aligns with its true density. The third scenario is an intermediate state where the internal channels are only partially occupied by polymer macromolecules.

The most complicated case is the third one. Equations (2)–(4) were used to determine the degree of intraphase, or the proportion of spaces filled by the polymer, from density data. Figure 8’s lower right panel displays this as a function of filler weight content. The findings show that LA has a greater ability than BER to pass through the CAH microparticle channels.

The final aspect ratio of the microparticles was affected differently by LA and BER, and they also exhibited varied propensities to form an intraphase. Rheological measurements were made since melt viscosity can affect both of these occurrences.

For each fabricated specimen, the dynamic viscosity versus oscillatory rate is illustrated in Figure 9. The neat LA polymer exhibited lower viscosity compared to the neat BER. Throughout the entire range of oscillation rates, BER demonstrated non-Newtonian fluid characteristics, while LA displayed flow independence at minimal oscillation rates, transitioning to pseudo-plasticity at elevated rates. Significantly, yield stresses were detected at low oscillation rates in the LA-based biocomposites, suggesting rheological jamming induced by solid–solid contacts between the microparticles [39].

This rheological behavior aligns with the observation that LA-CAH systems exhibit higher intraphase than their BER-based counterparts. At high frequencies, the lower complex viscosities of the LA biocomposites, relative to the neat polymer, suggest polymer degradation during processing. In contrast, BER-based composites maintained non-Newtonian flow behavior, similar to neat BER, indicating the matrix dominates the rheology of these systems.

Viscosity generally rose with reinforcement loading, notably with the CAH-B macroparticles (i.e., larger reinforcement dimensions). Conversely, the incorporation of CAH-A particulates surprisingly reduced the viscosity, dropping marginally beneath the pure BER system’s viscosity in the BER-10A specimen, specifically at elevated oscillation rates. This reduction in melt viscosity, particularly under high shear conditions, often suggests polymer breakdown.

To confirm this, we measured the proportional viscosity for the recovered polymers in a THF solution using a capillary viscometer (Figure 10). The resulting data indicate that incorporating CAH-B and CAH-A substantially diminished the efflux periods of LA separated from the biocomposites, whereas this influence was less notable in BER. This leads us to hypothesize a key difference between the two polymers: the highly viscous BER was more effective at physically shredding the CAH particles, which reduced their aspect ratio (A_f_). Conversely, the shrinkage effect was less intense in LA, as its degradation decreased viscosity. The greater intraphase formation in LA is likely a result of both this degradation and its slightly higher hydrophilicity. At the highest loading levels, the differences between LA and BER diminish. According to Vidal et al. [40], this may be because BER’s stronger shrinkage of the fillers promotes an increase in the intraphase, thereby offsetting the initial differences.

The stress–strain curves in Figure 11 represent the mechanical properties of the composites. This figure displays the representative curves for LA-based composites in Figure 11A and BER-based composites in Figure 11B. The insets of each figure provide a magnified view of the low-strain region, allowing for a more detailed analysis of the initial deformation behavior.

The mechanical properties of the composites—elastic modulus (E), tensile strength (TS), elongation at break (EB), and toughness—as a function of filler content are presented in Figure 12A–D. The addition of CAHs had a predictable stiffening effect, with all composites exhibiting higher elastic moduli than their corresponding neat polymers. However, this stiffening came at the cost of other properties; both tensile strength and, particularly, elongation at break generally worsened after the fillers were added. This behavior has been previously documented for similar composite systems [41].

The stiffness improvement was most notable in BER-based systems, which, as Tremeac et al. [41] noted, is attributed to the higher stiffness contrast between the CAH filler and the BER matrix than the LA system. The neat polymer moduli were 1.89 (±0.03) GPa for LA and 122 (±2) MPa for BER, while the bulk modulus of the CAH particles is approximately 13 GPa. The resulting composite modulus is also influenced by the degree of intraphase formation, which varies between the two polymers and affects the overall sample porosity [42].

A separate analysis of toughness revealed a dramatic decrease in all composites compared to their neat matrices. The highest toughness values were observed in BER-based composites (BER-10A > BER-10B > BER-20A > BER-20B). LA-based composites, however, were notably brittle, likely due to LA’s inherent fragility and degradation during melt processing. We hypothesize that the empty CAH channels may act as stress concentrators, leading to premature failure and lower ductility for a rigid filler system.

In order to compare the two matrices and evaluate the reinforcing impact, the dimensionless elastic modulus (Ec/Ew) was calculated. Three parameters were analyzed using a 2^3^ complete factorial design: polymer type (LA and BER), filler size (CAH-A and CAH-B), and filler amount (10% and 20%). These parameters’ importance was evaluated via Analysis of Variance (ANOVA), and the resulting data are presented concisely in Table 4.

Data dispersion for each experimental condition was assessed by analyzing the range of dimensionless moduli. Salmins et al. [43] define the minimum value as the quotient of the lowest composite modulus and the highest matrix modulus, and the maximum value as the quotient of the highest composite modulus and the lowest matrix modulus. BER-based materials displayed negligible data dispersion, with minimal divergence between a sample’s highest and lowest values. In sharp contrast, LA-based exhibited a greater spread between their minimum and maximum values.

Using Equation (9), the variance of the effect, V(E), was calculated as 0.000373, which gave a standard error (SE) of 0.019307. A *t*-test indicated that only main effects exceeding the threshold of (0.019307 × 2.3) = 0.044407 were statistically significant.

The symbols “+” and “−” are used to indicate the higher and lower levels of the variables, respectively, in the matrix of contrasts in Table 5. Using this matrix, we were able to use Equation (8) to determine the variables’ primary effects and interactions with one another, as seen in the table’s final row. We could determine which variations were statistically significant and which were probably the consequence of experimental noise by comparing these effects to the statistical significance threshold obtained by ANOVA (again, see the last row of Table 3).

All factors and their interdependencies were deemed statistically significant in this analysis. Figure 13A furnishes a visual representation of the principal impacts of the three parameters on the dimensionless elastic modulus (E_c_/E_w_).

The reinforcement content ranked as the second most influential factor, subsequent to the matrix classification. This characteristic was minimally impacted by the reinforcement’s aspect ratio, likely resulting from reinforcement scission during compounding, which diminished the final aspect ratio (Af). The pairwise relationships between reinforcement level and reinforcement dimension (B), reinforcement content and matrix classification (C), and reinforcement size and matrix type (D) are also charted in Figure 13.

Binary interaction plots, shown in Figure 13B–D, illustrate the relationship between the dimensionless elastic modulus (EC/EM) and each pair of variables. These plots help detect or predict a trend inversion at the intersection of the two lines. Specifically, the filler content–filler size interaction (Figure 13B) predicts an intersection at approximately 45% filler weight content. This suggests that at filler loadings above 45%, composites with CAH-A will exhibit a greater stiffening effect than those with CAH-B. This phenomenon may be attributed to the greater tendency of CAH-A to form an intraphase, resulting in less porous structures.

The remaining two interactions, filler content–polymer type (Figure 13C) and filler size–polymer type (Figure 13D), hold no practical significance. The intersection for the former is predicted at a filler content of $\phi_{w}$ = 0. In contrast, the latter’s intersection is projected at a physically and technologically irrelevant negative value for filler size. Despite the system’s complexity, which involves strong interrelations among interphase, intraphase, viscosity, aspect ratio, and degradation, the full factorial design effectively describes the behavior of the dimensionless modulus.

The dimensionless toughness was determined by standardizing the experimental result of each specimen against its corresponding neat matrix benchmark to investigate the influence of CAH concentration and size on the toughness of blends prepared using distinct polymeric systems. A full factorial design was then utilized to assess this metric. Reinforcement dimension (CAH-A or CAH-B), matrix classification (LA or BER), and reinforcement loading level (10% or 20%) were the factors examined. As shown in Table 6, an initial ANOVA was executed. The last row of Table 7’s contrast matrix presents the core impacts of each parameter and their interdependence, with statistically significant variables and interactions indicated.

All factors are significant when compared to the statistical significance level in Table 6 for the main effects and interactions (see the last row of Table 7). These factors’ significant rankings align with the findings for the dimensionless elastic modulus (EC/EM): filler aspect ratio > filler content > polymer kind.

But there was a significant shift in the primary impacts’ sign. Every major effect was negative, suggesting that the toughness of the neat polymers was adversely affected by the addition of CAH. The graphical depiction of the primary consequences in Figure 14A provides a good illustration of this.

We can speculate that the type of polymer has the biggest impact on this result. This is probably because there is a larger ductility disparity between the rigid CAH and the extremely stretchy BER than there is between the brittle LA and CAH.

Toughness may also be influenced by the intraphase proportion. Our investigation indicated that the BER blends exhibited a diminished intraphase ratio. The unoccupied voids of the CAH may have functioned as stress amplification sites, resulting in premature failure of the BER-based specimens. Discontinuity locations and the volume fraction of channel pores also increase commensurate with the reinforcement loading level. However, because the aspect ratios of CAH-A and CAH-B showed negligible deviation following compounding, the influence of the reinforcement aspect ratio was less observable. The greater intraphase detected in CAH-A may have partly offset the effect of a reduced particulate dimension, which would typically augment stress concentration locations.

According to the analysis of pairwise relationships, just the interdependence between reinforcement aspect ratio and matrix classification was deemed statistically insignificant. Figure 14B–D exhibits the binary interaction maps, which clarify whether a “mathematical” correlation holds any practical relevance. In this instance, the variables of practical interest were not expected to intersect. The intersection of the reinforcement level–reinforcement dimension relationship, which held the greatest statistical importance, was projected at diminished dimensionless toughness readings (Figure 14B). Conversely, the intersection for the reinforcement level–matrix classification relationship was anticipated at negative reinforcement content values (Figure 14C). Therefore, a straightforward analysis of principal effects can sufficiently characterize the toughness response.

## 4. Conclusions

The feasibility of employing CAHs as reinforcement for polylactic acid (LA) and a bio-based epoxy resin (BER) was investigated herein. The influences of matrix classification, reinforcement aspect ratio, and loading level on the ultimate toughness and elastic modulus of the resulting blends were statistically assessed. Furthermore, we scrutinized the correlation between the reinforcements’ final aspect ratio, degradation, intraphase development, and viscosity.

Because of the greater stiffness disparity between the reinforcement and the matrix, the stiffening effect of incorporating CAHs was more pronounced in BER blends. Matrix classification had the most influence on the dimensionless elastic modulus, with the reinforcement aspect ratio having the least. This is attributed to the reinforcements’ scission during compounding, which diminishes the aspect ratio. The complete factorial analysis exposed a substantial interdependence among reinforcement dimension and loading level due to the system’s complexity. Consequently, CAH-B was forecast to be a superior reinforcing agent compared to CAH-A up to a loading level of 45%, beyond which CAH-A would impart greater rigidity.

We observed the identical hierarchy of importance for dimensionless toughness (matrix classification > reinforcement content > reinforcement aspect ratio). However, in this case, all principal effects were negative, suggesting that the incorporation of CAH lowered the durability of the neat polymers.

## Figures and Tables

**Figure 1 polymers-17-03013-f001:**
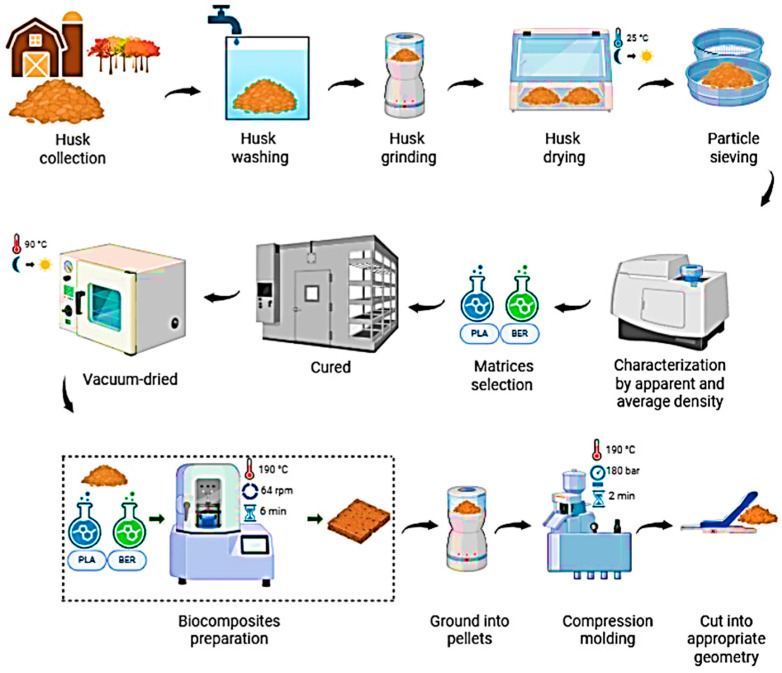
Materials and preparation.

**Figure 2 polymers-17-03013-f002:**
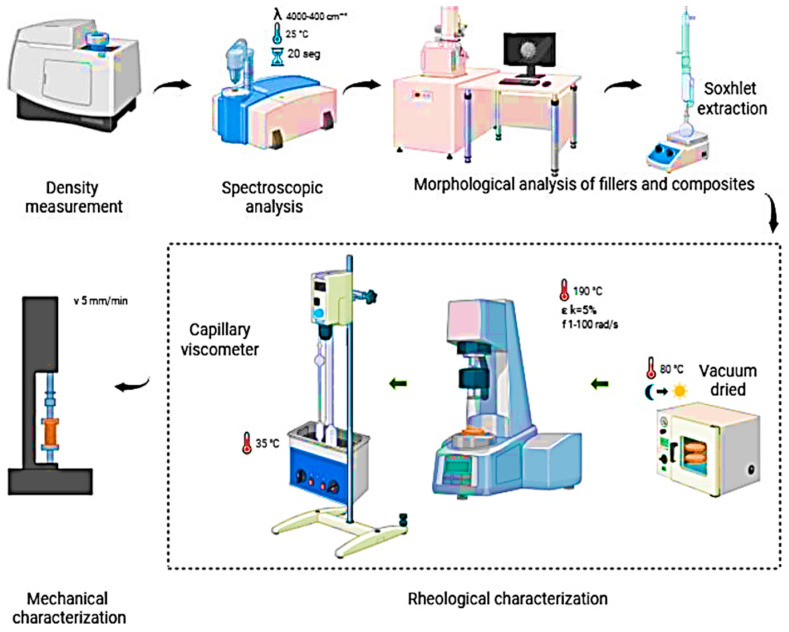
Characterization techniques.

**Figure 3 polymers-17-03013-f003:**
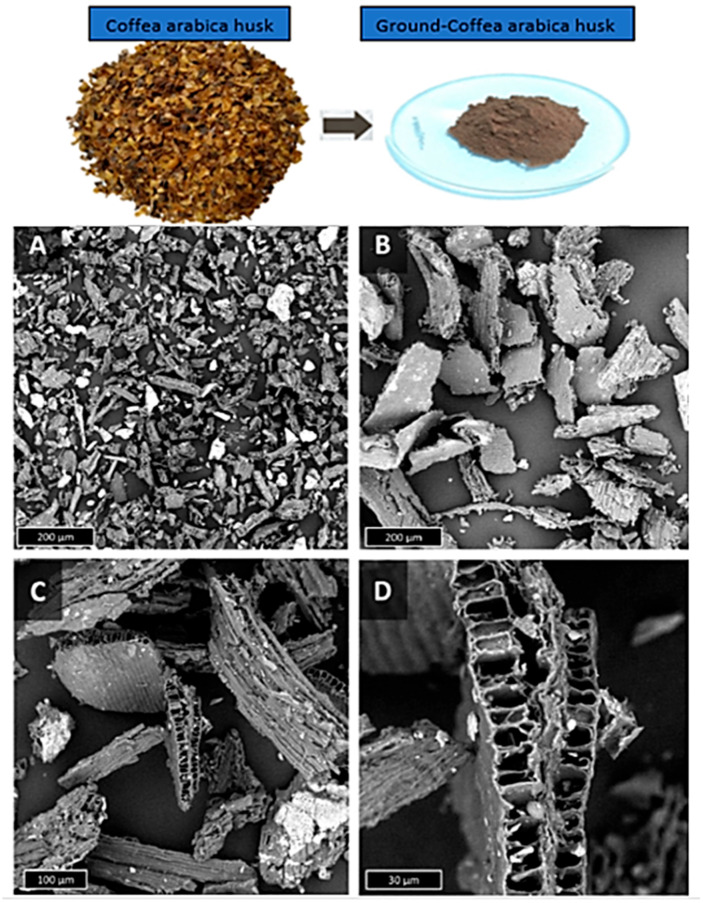
Raw Coffea arabica husks captured in digital images. CAH-A (**A**) and CAH-B (**B**) SEM micrographs, as well as CAH-B detailed micrographs (**C**,**D**).

**Figure 4 polymers-17-03013-f004:**
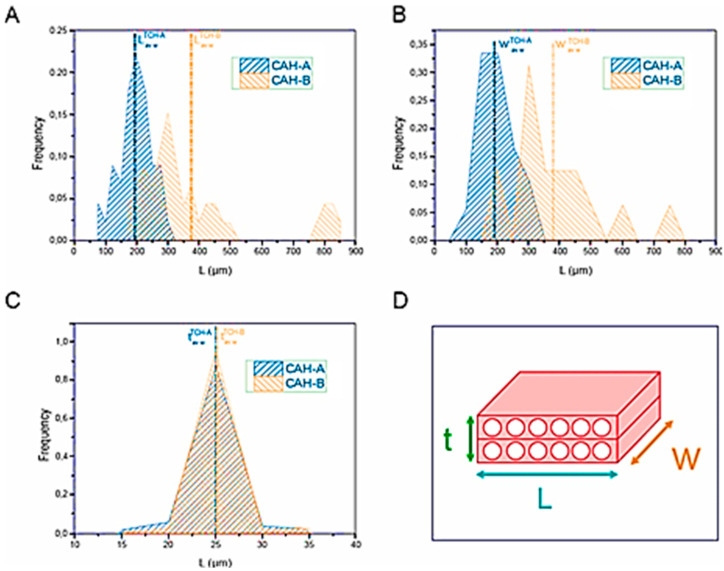
Distributions of CAH-A and CAH-B’s length (**A**), width (**B**), and thickness (**C**), as well as their average weight values, are shown graphically in the CAH microparticle’s geometrical parameters (**D**).

**Figure 5 polymers-17-03013-f005:**
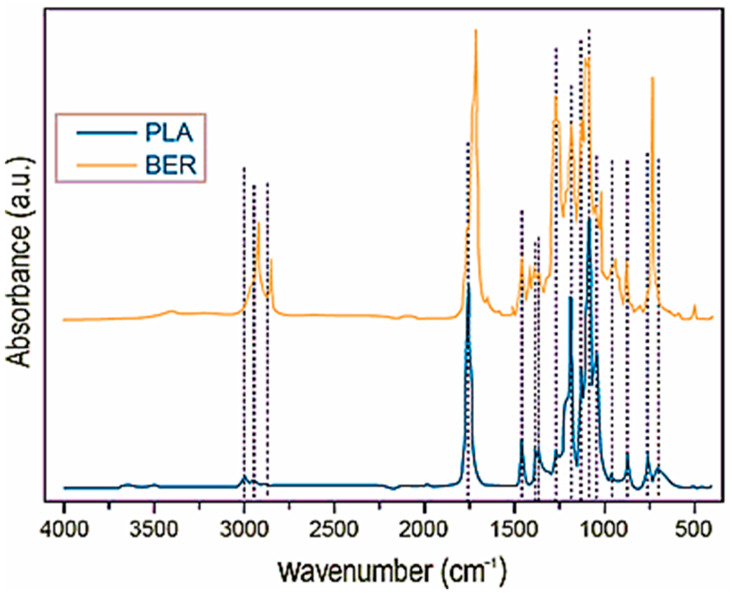
The FTIR/ATR spectroscopic data for BER (top) and LA (bottom).

**Figure 6 polymers-17-03013-f006:**
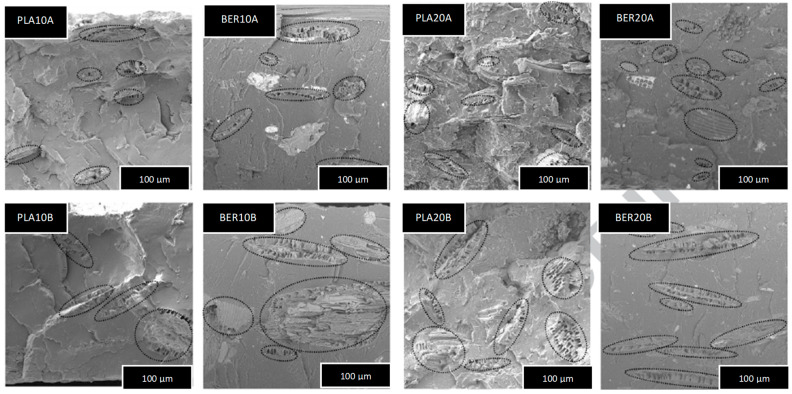
SEM examination of the eight samples under investigation’s cryofractured cross-sections. Magnification state 100 μm.

**Figure 7 polymers-17-03013-f007:**
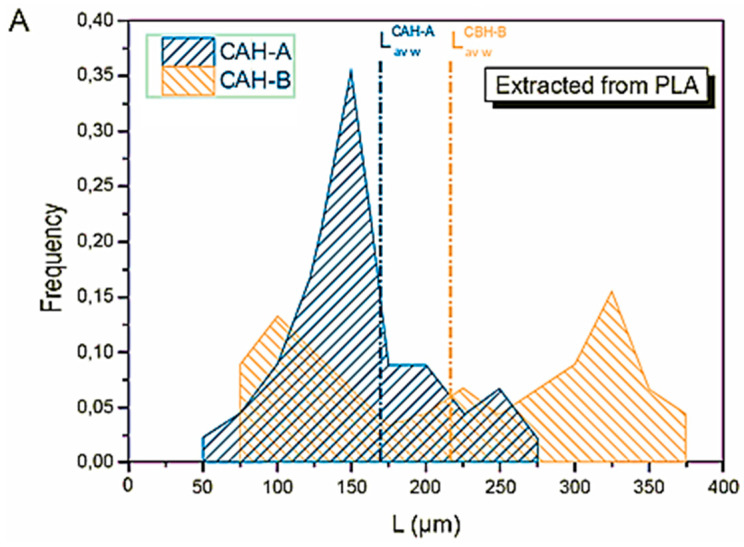
**(A**) CAH-A and CAH-B’s length distribution following mixing with LA (**B**) CAH-A and CAH-B’s length distribution following BER mixing (**C**) the impact of melt-mixing time on CAH-A and CAH-B’s aspect ratio.

**Figure 8 polymers-17-03013-f008:**
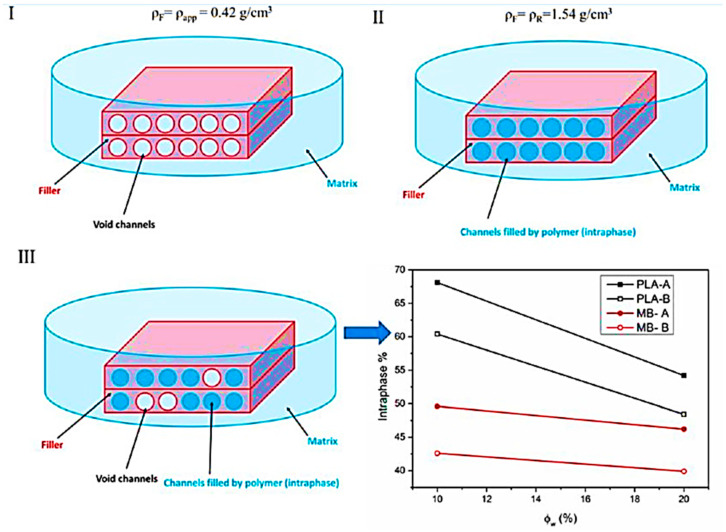
A schematic illustration detailing the CAH reinforcements with unoccupied (**I**), saturated (**II**), and partially occupied (**III**) voids is provided. This is accompanied by the intraphase percentage graphed against the reinforcement loading level for the material blends under investigation (**bottom right**).

**Figure 9 polymers-17-03013-f009:**
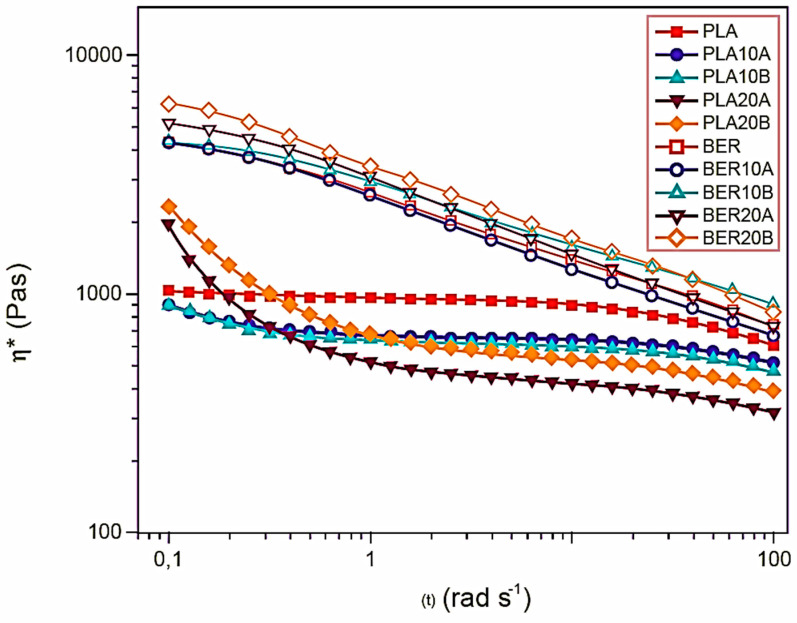
The systems’ melt rheology curves.

**Figure 10 polymers-17-03013-f010:**
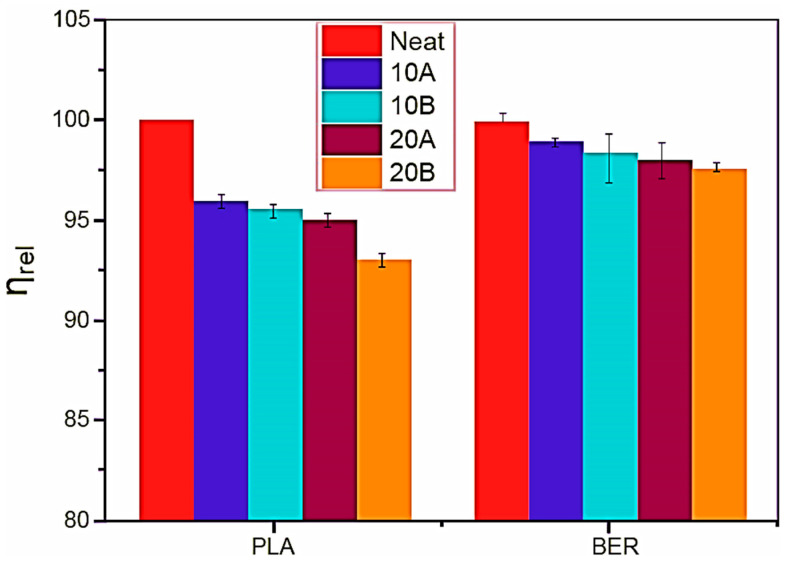
After filler removal, the relative viscosity of polymer solution in THF for each system.

**Figure 11 polymers-17-03013-f011:**
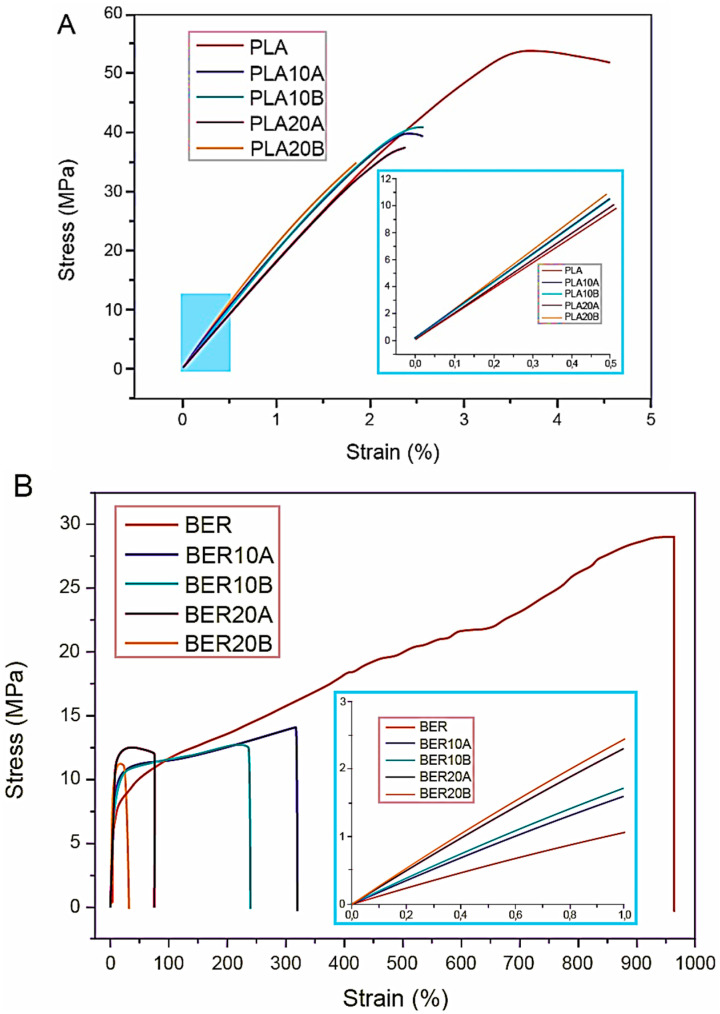
The inset of each image contains characteristic stress–strain plots for (**A**) LA-based blends and (**B**) BER-based blends, in addition to the plot’s initial gradient.

**Figure 12 polymers-17-03013-f012:**
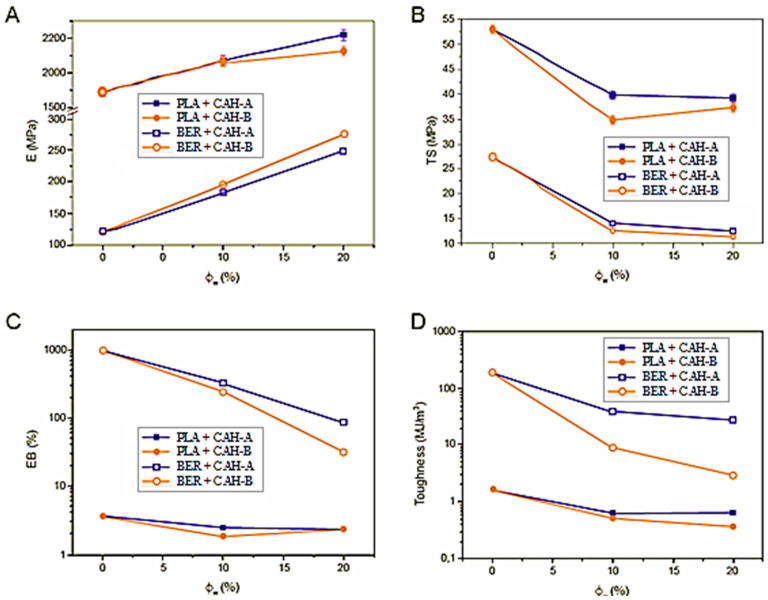
Plotting of toughness (**D**), EB (**C**), TS (**B**), and E (**A**) against filler weight content.

**Figure 13 polymers-17-03013-f013:**
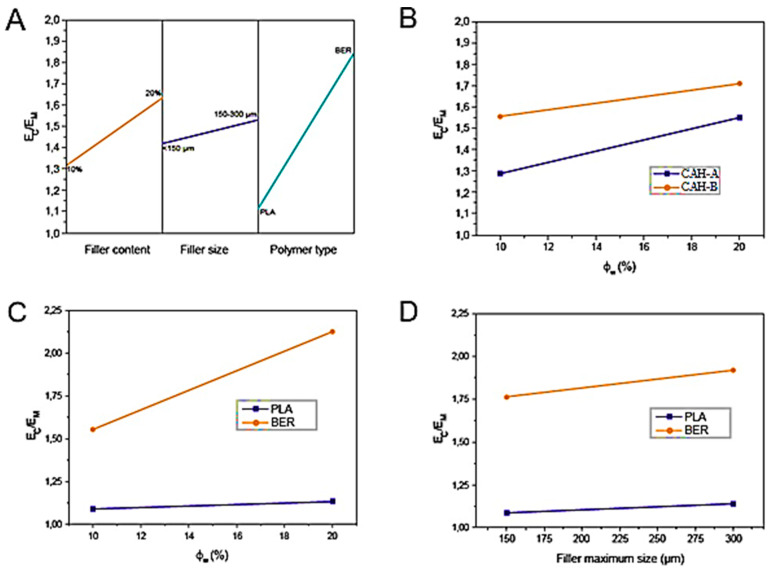
A graphic interpretation of the principal influence of matrix classification, reinforcement magnitude, and reinforcement level on EC/EM is presented in panel (**A**). The pairwise relationships between reinforcement content and reinforcement dimension (**D**), reinforcement content and matrix classification (**C**), and reinforcement content and reinforcement dimension (**B**) are charted via a map.

**Figure 14 polymers-17-03013-f014:**
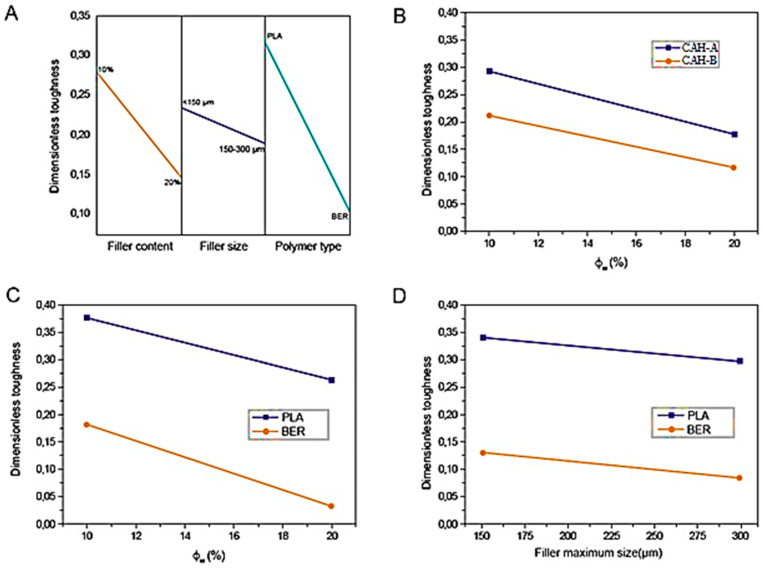
The principal influence of matrix classification, reinforcement dimension, and reinforcement level on dimensionless toughness is charted in panel (**A**). The pairwise relationships between reinforcement content and reinforcement magnitude (**D**), reinforcement content and matrix classification (**C**), and reinforcement dimension and matrix classification (**B**) are illustrated via a map.

**Table 1 polymers-17-03013-t001:** Key characteristics of the polymeric systems employed.

Sample	Density (g/cm^3^)	MFI (g/10 min)	T_m_ (°C)	Molecular Weight (Da)
LA	1.3	6	149	Mn = 112,990
BER	1.2	~8	≈90	Mn = 8000

**Table 2 polymers-17-03013-t002:** Sample formulation under investigation.

Reference Symbol	Polymeric Media	CAH Proportion (wt %)	CAH Particle Granularity (μm)
LA	LA2002D	-	
LA10A	LA2002D	15	74–149
LA20A	LA2002D	30	74–149
LA10B	LA2002D	15	149–299
LA20B	LA2002D	30	149–299
BER	BER170G	-	
BER10A	BER170G	15	74–149
BER20A	BER170G	30	74–149
BER10B	BER170G	15	149–299
BER20B	BER170G	30	149–299

**Table 3 polymers-17-03013-t003:** Important CAH-A and CAH-B geometrical parameters prior to melt compounding approach.

CAHs Category	L	W	t	$D_{eq}$	$A_{f}$
A	196.2	99.6	26	154.8	7.1
B	375.2	198.5	26	311.7	13.3

**Table 4 polymers-17-03013-t004:** Variance Assessment for E_c_/E_w_.

Experimental Runs	1	2	3	Data Dispersion		
	% CAH	CAH Particle Dimension	Polymer Dimension	Min	Max	Mean
LA10A	−	−	−	1.051	1.137	1.087
LA20A	+	−	−	1.053	1.131	1.092
LA10B	−	+	−	1.072	1.123	1.097
LA20B	+	+	−	1.134	1.235	1.188
BER10A	−	−	+	1.517	1.501	1.506
BER20A	+	−	+	1.986	2.058	2.023
BER10B	−	+	+	1.614	1.611	1.605
BER20B	+	+	+	2.238	2.242	2.240
V(E)				0.0003		
SE				0.019		
Significance threshold				0.044		

**Table 5 polymers-17-03013-t005:** The core influences and two-way relationships among the experimental parameters on Ec/Ew are derived from the contrast matrix (1 = filler content, 2 = reinforcement dimension, 3 = matrix classification).

Experimental Runs	Main Effects			Binary Interactions			E_c_/E_w_
	1	2	3	1–2	1–3	2–3	
LA10A	−	−	−	+	+	+	1.086
LA20A	+	−	−	−	−	+	1.089
LA10B	−	+	−	−	+	−	1.095
LA20B	+	+	−	+	−	−	1.185
BER10A	−	−	+	+	−	−	1.499
BER20A	+	−	+	−	+	−	2.021
BER10B	−	+	+	−	−	+	1.604
BER20B	+	+	+	+	+	+	2.238
	0.311	0.106	0.726	0.108	0.265	0.054	1.477

**Table 6 polymers-17-03013-t006:** Dimensionless toughness ANOVA. 1 stands for filler content, 2 for filler size, and 3 for polymer kind.

Experimental Runs	1	2	3	Data Scattering		
	% CAH	CAH Size	Polymer Size	Min	Max	Mean
LA10A	−	−	−	0.37041	0.37924	0.37482
LA20A	+	−	−	0.29968	0.31257	0.30612
LA10B	−	+	−	0.36553	0.38553	0.37553
LA20B	+	+	−	0.21908	0.22075	0.21992
BER10A	−	−	+	0.20658	0.21588	0.21123
BER20A	+	−	+	0.04694	0.04905	0.04800
BER10B	−	+	+	0.14690	0.15350	0.15020
BER20B	+	+	+	0.01507	0.01575	0.01541
V(E)				9.778 × 10^−6^		
SE				0.00312		
Significance threshold				0.00719		

**Table 7 polymers-17-03013-t007:** Dimensionless toughness was examined using a matrix of contrasts to identify the primary impacts and binary interactions between the variables (1 being filler content, 2 being filler size, and 3 being polymer type).

Experimental Runs	Main Effects			Binary Interactions			Dimensionless Toughness
	1	2	3	1–2	1–3	2–3	
LA10A	−	−	−	+	+	+	0.37482
LA20A	+	−	−	−	−	+	0.30612
LA10B	−	+	−	−	+	−	0.37553
LA20B	+	+	−	+	−	−	0.21992
BER10A	−	−	+	+	−	−	0.21123
BER20A	+	−	+	−	+	−	0.04800
BER10B	−	+	+	−	−	+	0.15020
BER20B	+	+	+	+	+	+	0.01541
	−0.13058	−0.04477	−0.21288	−0.01911	−0.01844	−0.00201	0.21265

## Data Availability

The original contributions presented in this study are included in the article. Further inquiries can be directed to the corresponding author.

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
